

# Fundamental niche unfilling and potential invasion risk of the slider turtle *Trachemys scripta*

Sayra Espindola[1,2], Juan L. Parra[3] and Ella Vázquez-Domínguez[1,4]

[1] Departamento de Ecología de la Biodiversidad, Instituto de Ecología, Universidad Nacional Autónoma de México, Ciudad Universitaria, Ciudad de México, México
[2] Posgrado en Ciencias Biológicas, Universidad Nacional Autónoma de México, Coyoacán, Ciudad de México, México
[3] Grupo de Ecología y Evolución de Vertebrados, Instituto de Biología, Universidad de Antioquia, Medellín, Colombia
[4] American Museum of Natural History, New York, NY, United States of America

Corresponding author
Ella Vázquez-Domínguez,
evazquez@ecologia.unam.mx

## ABSTRACT

**Background**. How species colonize new environments is still a fundamental question in ecology and evolution, assessable by evaluating range characteristics of invasive species. Here we propose a model approach to evaluate environmental conditions and species features to predict niche changes in non-equilibrium contexts. It incorporates potentially range-limiting processes (fundamental niche), hence allowing for better predictions of range shifts, differentiation of analog and non-analog conditions between the native and non-native (invaded) ranges, and identification of environmental conditions not currently available but likely in the future. We apply our approach with the worldwide invasive slider-turtle *Trachemys scripta*.

**Methods**. We estimated the native and non-native realized niches of *T. scripta* and built its fundamental niche based on key features of the turtle's temperature physiological tolerance limits and survival-associated factors. We next estimated response functions adjusted to the physiological predictor variables and estimated habitat suitability values, followed by a comprehensive set of analyses and simulations to compare the environmental conditions occupied by *T. scripta* (at its native and non-native ranges).

**Results**. Climatic space analysis showed that the *T. scripta*'s non-native realized niche is 28.6% greater than the native one. Response curves showed that it does not use its entire range of temperature tolerances (density curves for native: 5.3–23.7 °C and non-native: 1.7–28.4 °C ranges). Whether considering the mean temperature of the warmest or the coldest quarter, it occupies a wider range of temperatures along its non-native distribution. Results of the response curves for worldwide (global) and across Mexico (regional) comparisons showed it occupies analog and non-analog conditions between its native and invaded ranges, exhibiting also unoccupied suitable climatic conditions.

**Discussion**. We demonstrate that *T. scripta* occupies a wider subset of its fundamental niche along its non-native range (within its physiological tolerances), revealing that the species observed niche shift corresponds to a different subset of its fundamental niche (niche unfilling). We also identified suitable environmental conditions, globally and regionally, where the slider turtle could potentially invade. Our approach allows to accurately predict niche changes in novel or non-equilibrium contexts, which can improve our understanding about ecological aspects and geographic range boundaries in current and potential invasions.

## INTRODUCTION

A basic aim of ecology and biogeography is to understand the distribution and abundance of organisms. Additionally, how species colonize new environments is still a fundamental question in ecology and evolution (*Hastings et al., 2005*; *Kueffer, Pyšek & Richardson, 2013*). Biological invasions offer a unique opportunity to explore the patterns and processes of species colonization into novel environments, also a key aspect for monitoring biotic exchange (*Rodríguez-Labajos, Binimelis & Monterroso, 2009*; *Banks et al., 2014*). Indeed, detecting areas where exotic species are likely to establish is a critical challenge for invasion ecology and biodiversity conservation, which in turn can aid in the evaluation and prevention of invasion risk (*Broennimann & Guisan, 2008*; *Higgins & Richardson, 2014*).

Different approaches have been applied to understand the features that facilitate the establishment of invasive species into new environments (*Kueffer, Pyšek & Richardson, 2013*), emphasizing the suitability of the abiotic features of the colonized habitat as a key prerequisite for their success (*Kueffer, Pyšek & Richardson, 2013*; *Colangelo et al., 2017*). Correlative ecological niche models (ENMs) have become a common tool to characterize the environmental conditions suitable for invasive species (*Peterson, 2003*; *Jeschke & Strayer, 2008*). ENMs are based on correlations between the species occurrence distribution and environmental data—the realized niche—(see Glossary S1). However, whether species retain their realized niche when introduced elsewhere is an unresolved ecological query, for which findings are largely diverse and sometimes controversial (*Early & Sax, 2014*; *Guisan et al., 2014*), while evidence supports both niche conservatism and niche shifts during invasions (*Petitpierre et al., 2012*; *Guisan et al., 2014*). In fact, *Early & Sax (2014)* contrasted the climatic conditions occupied by 51 plant species in Europe (native) and USA (non-native) distributions and found that a large proportion of the latter distributions occurred outside the climatic conditions occupied in their native ranges. Also, more than half of the invaded ranges of 71 reptiles and amphibians showed niche shifts (*Li et al., 2014*), while *Strubbe, Beauchard & Matthysen (2015)* showed, for an evaluation combining 29 vertebrate species, that niche overlap between native and non-native populations was generally low because of a large degree of niche unfilling (see Glossary S1) in the non-native range.

Evaluating these shifts under the framework proposed by *Guisan et al. (2014)*, which takes into account occurrence data of the realized climatic niche at the native and non-native ranges, is a first step in understanding whether the species' niche is conserved or if it can undergo shifts in the novel environment. It also aids in the assessment of whether the changes detected are likely caused by native niche unfilling in the non-native range, or by expansion into novel environments (*Broennimann & Guisan, 2008*; *Tingley et al., 2014*). However, it is desirable to explicitly incorporate potentially range-limiting processes, namely physiological tolerances and constraints of organisms. Incorporating links between

the functional traits of organisms and their environments provides a mechanistic view of Hutchinson's fundamental niche (*Hutchinson, 1957*; *Hutchinson, 1978*) (see Glossary S1), which can then be mapped to the landscape to infer range shifts (*Kearney et al., 2008*; *Kearney & Porter, 2009*; *Rodrigues, Coelho & Ribeiro, 2018*). Furthermore, in order to model a species' niche mechanistically and infer its potential range, the organism's data must be considered in the model not as a point on a map but rather as a set of traits (e.g., morphological, behavioural, physiological; see a detailed explanation in *Kearney & Porter, 2009* and their Fig. 1). Also, adding range-limiting processes to ENMs allows to differentiate analog and non-analog conditions (Glossary S1) between the native and invaded ranges, and to identify environmental conditions that are not currently available but could be in the future (*Jackson & Overpeck, 2000*). However, studies contrasting fundamental and realized niche models in shaping the geographic range limits and niche changes are scarce (but see *Soberón & Arroyo-Peña, 2017* for a meta-analysis; *Rodrigues, Coelho & Ribeiro, 2018* and *Allen-Ankins & Stoffels, 2017* for analyses of thermal niches). Much less has been investigated regarding invasive species (*Rödder et al., 2009*; *Tingley et al., 2014*), likely as a result of limited information on physiological responses, thus attempts to estimate niche changes could render erroneous conclusions.

*Trachemys scripta elegans* is considered one of the worst invasive species in the world (*Lowe et al., 2000*), which has been studied mainly about its extraordinary potential for impacting indigenous habitats and species (e.g., *Cadi & Joly, 2004*; *Cadi et al., 2004*; *Ficetola, Thuiller & Padoa-Schioppa, 2009*). Previous studies using species distribution models to explore the invasion patterns of *T.s. elegans* have shown that reproductive populations (in central and northern Italy) are associated to warmer climates compared with its native range (*Ficetola, Thuiller & Padoa-Schioppa, 2009*), while *Rödder et al. (2009)* suggested that climatic requirements during egg incubation could be a major driver for the species' native geographic distribution. Recent studies showed that the invasion process of this turtle species has involved niche shifts (*Li et al., 2014*; *Rodrigues et al., 2016*), and conclude that a dynamic method to properly predict its potential invasion risk is still lacking (*Rodrigues et al., 2016*). Moreover, it is still unclear whether *T.s. elegans* is filling its fundamental niche or if, alternatively, there are unoccupied areas that can potentially be invaded.

We here present a model approach for characterizing potential environmental conditions and species features in novel or non-equilibrium contexts, like species invasions (see Fig. 1). It further advances what *Jackson & Overpeck (2000)* and *Guisan et al. (2014)* have proposed, namely to represent the differences among realized and fundamental niches, as well as the different niche changes that can be observed based on the occupied ranges and within analog conditions. Models based on physiological tolerances data or other fundamental traits (i.e., mechanistic models) are independent of the species' current distribution, hence providing a more accurate prediction of where a species can survive and reproduce in the absence of biotic interactions and dispersal limitations (*Soberón, 2007*; *Kearney & Porter, 2009*; *Tingley et al., 2014*). We hereafter refer to the fundamental niche as that of *Soberón & Arroyo-Peña (2017)*: "the fundamental niche (NF) of a species is determined by its physiological range of tolerance to environmental factors in the absence of biotic interactions, whereas the regions of the planet with environments in NF would represent
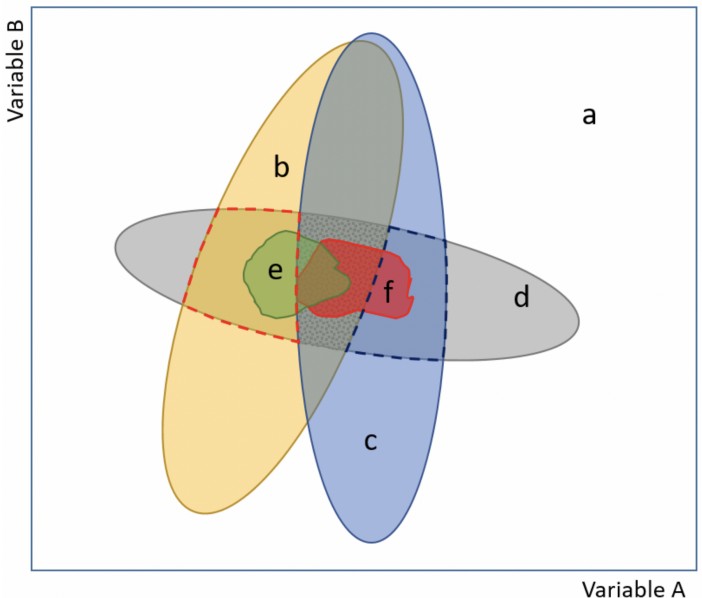

**Figure 1 Scheme of our proposed model approach for evaluating potential environmental conditions and niche changes for species invasions.** The model explicitly incorporates information about the realized and the fundamental niches (see Glossary S1), and also encompasses both analog and non-analog climate (basic figure based on (*Jackson & Overpeck, 2000*; *Guisan et al., 2014*). The combination of two variables (axes A and B) depicts the environmental space (a), (b) and (c) correspond to the current environment available at the native and the invaded (non-native) ranges, respectively, while (d) delimits the fundamental niche (i.e., areas based on range-limiting processes, like physiological tolerance limits). Importantly, notice that there are areas in (d) that are not currently available, but are areas of potential distribution if an environmental change occurs. The realized (occupied) niche is represented by (e) at the native range and (f) at the invaded range (i.e., areas invaded but outside of the native distribution). Indices of niche shift (unfilling, stability and expansion; see Glossary S1), as described in *Guisan et al. (2014)*, occur inside the central dotted grey area, which corresponds to the analog conditions between the native and invaded ranges. Non-analog conditions but comprising a potential range to be invaded, because it is within the fundamental niche, are indicated with red (native range) and blue (invaded range) broken lines.

some sort of potential area of distribution for the species''. Accordingly, we describe how to combine into one approach the realized and fundamental niches, encompassing both analog and non-analog conditions, to clearly evaluate if the invaded range represents a niche change but also to adequately define what kind of change it refers to (unfilling, expansion, stability; see Glossary S1), and to identify environmental conditions that are not currently available but could be in the future (*Jackson & Overpeck, 2000*). Importantly, we incorporate a means to identify non-analog regions (mostly absent in invasive species distribution models), which ultimately comprise a potential area of distribution that can be invaded because it is within the species' fundamental niche range (Fig. 1).

Hence, here we follow our proposed model to assess the current worldwide environmental conditions and distribution ranges of the slider turtle. To this end, we performed a comprehensive set of analyses to compare native and non-native realized and fundamental niches, jointly with estimations of response functions adjusted to the physiological predictor variables, including simulations and response curves, to obtain

habitat suitability values. We also test our model regionally, by estimating the potential distribution (invasion risk) for Mexico. Our premise was that we would be able to describe this turtle's niche change patterns throughout its invasion, and resolve whether it is filling its fundamental niche or can invade new environments. Based on our results, we also aimed to exemplify that a more realistic model to define potential areas of invasion risk can be performed based on indirect (i.e., not experimentally based) fundamental traits.

## MATERIALS & METHODS

### Occurrence and climatic data

Considering the overlapped distribution of *Trachemys scripta* subspecies (*T.s. scripta*, *T.s. elegans* and *T.s. troostii*), and the lack of formal discrimination of the turtle's invasive introductions on the basis of subspecies (*Van Dijk, Harding & Hammerson, 2011*), we obtained georeferenced records for the three subspecies from the Global Biodiversity Information Facility (*GBIF, 2017*) and the *VertNet (2016)* databases. To complement the latter, we also included our own collecting records and those from a thorough revision from the available published literature (data available at https://figshare.com/articles/_/8175158). In order to define *T. scripta*'s native range that encompasses from eastern and central United States of America to extreme north-eastern Mexico, we used as a basis the range maps proposed by *Seidel (2002)* about taxonomic observations on extant species and subspecies of *Trachemys*, updated in *Rhodin et al. (2017)*; next, we used the Freshwater Ecoregions of the World (*WWF/TNC, 2013*) to define range limits based on a global freshwater regionalization. Data points outside these limits were considered as non-native records; that is where the species has been introduced and is considered invasive worldwide (including North America where it has also been introduced outside its native range). We applied a geographic filter to our full set of occurrence data, considering records further than 5 km distance as independent points (multiple points within a cell indicate abundance is higher on those sites in comparison to cells with fewer occurrences). To that end, we used the Ecospat package in R (*Di Cola et al., 2017*) to remove duplicated points within a 5 km buffer, thus reducing overfitting and improving predictions (*De Oliveira et al., 2014*). As a result, we have the most comprehensive data set for the slider turtle, with 2,552 occurrence records, 1,395 from the native distribution range and 1,157 from non-native occurrences around the globe, more than four times the number of records from previous studies.

Climatic data were obtained from the global climate layers of WorldClim (*Fick & Hijmans, 2017*), which represents a statistical summary of temperature, precipitation and radiation at 5ʹspatial resolution. We included the six following bioclimatic variables: annual mean temperature (BIO1), mean temperature of the warmest quarter (BIO10), mean temperature of the coldest quarter (BIO11), annual precipitation (BIO12), precipitation of driest month (BIO14), and annual mean radiation (RAD), which were selected on the basis of the slider turtle's natural history (i.e., physiology), and also to reduced correlated variables (Pearson's correlation coefficients <0.85). *Trachemys* slider turtles are found in shallow, slow-moving water that has diverse vegetation and nearby places to bask. During seasonal dryness turtles may wander far seeking water sources, rapidly colonizing any

newly available habitat; hence, waterbodies availability is a key feature for the activity, dispersal, distribution, and overall life history and survival of this species (*Cox et al., 1998*; *Minton, 2001*; *Buhlmann, Tuberville & Gibbons, 2008*). Accordingly, we constructed a layer depicting the Euclidian distances to the nearest freshwater body (NearDist) (data available at https://figshare.com/articles/_/8175158). For this, we used the Global Lakes and Wetlands Database (GLWD; *WWF, 2017*), a combination of best available sources for lakes and wetlands on a global (world) scale. This raster map has a resolution of 30-second that includes polygons of great lakes (area $\geq$ 50 km$^2$), great reservoirs (capacity $\geq$ 0.5 km$^3$) and permanent water bodies (area $\geq$ 0.1 km$^2$). This is the best resolution available and, although smaller water bodies might not be directly detected, nonetheless polygons encompass the maximum extension of different kinds of water bodies (e.g., marshes, flooding areas). Importantly, this distance to the nearest freshwater body is an indirect measure of water availability—not a measure of dispersal—hence it defines a characteristic of the fundamental niche (*Anderson, Gutierrez & Romano, 2002*; *Braun & Phelps, 2016*).

We tested our proposed model at a regional scale, for Mexico, based on the same variables described above but with a higher resolution (30 arc sec). We used climatic surfaces for the average monthly climate period 1910–2009 (*Cuervo-Robayo et al., 2014*), jointly with a layer we built depicting the Euclidian distances to the nearest freshwater body for Mexico (NearDistMex) using the National Hydrographic System catalogue (*SGM, 2019*).

## Comparisons between native and non-native realized niches

We used different approaches to assess similarities and differences between the climatic conditions occupied by *T. scripta* in its native and non-native ranges, all performed in R 3.0.1 (*R Core Team, 2015*). First, in order to characterize the environments available worldwide, we used as reference the occurrence records of all freshwater turtles (not only *Trachemys*) from *GBIF (2017)*, which represent the environments that turtles could occupy. However, because some regions are better sampled for turtles than others and we cannot know if the lack of data is because the environment is not suitable or because they have not been sampled, we constructed a probability layer based on sampling effort by means of assigning a higher probability value where higher occurrences are recorded. We then used the DISMO package to create 100,000 random points worldwide weighted by our probability value. Next, based on the bioclimatic variables of those random points and of the 2,556 *T. scripta* occurrence records (data available at https://figshare.com/articles/_/8175158), we performed a weighted principal components analysis (PCA). Finally, using the first two axes of this PCA we: (*i*) draw contours estimated with a kernel density function (*Broennimann et al., 2012*) to delimit the climatic conditions available on the native range, the non-native ranges (in America, Europe, Asia and Australia) and around the world (areas that have not been invaded); and (*ii*) draw points depicting the estimated realized niche of the native and non-native ranges. This procedure randomizes the position of the kernel density surface of one of the two species within the environmental space available for it, allowing to evaluate whether the realized niches (kernel density surface) occupied in native and non-native ranges are more or less similar to the distribution of similarities under a null model.

As a complementary test, we compared climatic covariance matrices (native versus non-native ranges) with a Common Principal Components Analysis (CPCA; *Phillips & Arnold, 1999*), which compares two or more matrices considering their eigenvectors and eigenvalues in a hierarchical fashion, to describe their structure in relation to the size, shape and orientation of the matrices. We used the latter to establish principal components that were common between matrices (i.e., progressive differences in shape, orientation and size) and to test hypotheses about equality (identical eigenvectors and eigenvalues, i.e., identical size, shape and orientation), proportionality (equal eigenvectors, but eigenvalues differing in a scalar amount, i.e., same shape and orientation, different but proportional size), and unrelated structure (matrices have dissimilar eigenvectors and eigenvalues). The best solution under the model-building approach of the CPCA of the comparison of climatic covariance matrices is indicated by the minimum value of Akaike information criterion (AIC; *Phillips & Arnold, 1999*).

The statistical framework proposed by *Broennimann et al. (2012)* was used to evaluate the assumption of niche conservatism in biological invasions by quantifying the similarity between the native and the non-native realized niches. These comparisons involve a multivariate analysis to calibrate the niche and the occurrence density, performed with Ecospat, and the estimation of niche overlap using Schoener's $D$ metric (*Schoener, 1970*), an index ranging from 0 (no overlap) to 1 (total overlap) (*Warren, Glor & Turelli, 2008*; *Broennimann et al., 2012*), with statistical significance assessed on the basis of 100 randomizations ($\alpha = 0.05$). Furthermore, the density of occurrences in environmental space was used to estimate niche expansion (new environmental conditions found in the non-native range), stability (proportion of the native niche conditions found in the non-native one), and unfilling (proportion of the native niche not occupied in the non-native) (*Guisan et al., 2014*), with Ecospat. Finally, since the previous method only allows the use of two principal components, for a high-dimensional niche overlap analysis we used a technique based on a multivariate kernel density estimation, implemented in the Hypervolume in R (*Blonder et al., 2014*), that estimates densities based on a Monte Carlo importance sampling approach. To define the six-dimensional hypervolumes (i.e., the six bioclimatic variables described above) for the native and non-native ranges, we chose a threshold that included 100% of the total probability density. We then estimated the overlap (intersection) between the two hypervolumes (native and non-native), and the hypervolume unique to each native or non-native range; results are depicted as pair-plots.

## Environmental tolerances, fundamental niche and potential distribution

We performed a literature review (Table S1) to obtain information about physiological optimums and tolerance limits for *T. scripta*. The environmental variable reported more often and consistently was temperature for the following five features: during eggs incubation, hatching, growth, basking, and activity periods. The proximity to freshwater bodies is another critical limit, a fundamental factor for the turtles' survival which they depend upon to complete their life cycles, feeding and reproduction (*Anderson, Gutierrez & Romano, 2002*; *Braun & Phelps, 2016*); thus, we used the Euclidian distance to the nearest

freshwater body (NearDist) as an indirect measure of water availability, as explained above. We defined a 10 km buffer for distance to water availability, considering the general 5–10 km average distance reported where turtles can explore (*Cagle, 1944*) and, therefore, reach freshwater bodies. Hence, we considered the combined use of these different temperature tolerance limits and freshwater availability as an accurate approximation of the fundamental niche, as bivariate environmental space, for the slider turtle. Accordingly, we draw response curves in R by using beta and logistic functions adjusted to the optimal, maximum and minimum tolerance limits of *T. scripta*, with VIRTUALSPECIES in R (*Leroy et al., 2015*), which were then used to simulate its potential geographic distribution (see Glossary S1). The tolerance limits used, based on our literature review, were: optimum temperature for eggs incubation (29.5 °C) and daily activity (25.6 °C); ranges for feeding (16.8–32.1 °C), normal activity (10–37 °C), and basking (26.7–31.2 °C); and minimum and maximum tolerances before death (−12.6–42.3 °C). VIRTUALSPECIES allows to define response functions for different predictor variables and to combine the responses to obtain a habitat suitability value. This approach increases the ecological realism for generating environmental suitability of virtual species and to create more adequate species distribution models.

Next, to compare the range of the fundamental niche that is actually occupied by *T. scripta* with the simulated results, kernel density curves based on the temperature variables from all occurrence data (taking into account native and non-native records from which we extracted bioclimatic data) were overlapped with the simulated response curves. Contours and points were also drawn as described above (*Broennimann et al., 2012*), but specifically considering temperature data from the annual mean temperature (BIO1), the mean temperature of the warmest quarter (BIO10), and the mean temperature of the coldest quarter (BIO11), to encompass both the temperature ranges and the extreme tolerance limits of the five described activities; the variable NearDist was also included. The 100,000 generated worldwide random points were used to delimit the bivariate environmental space available, while the occurrence points were used to characterize the bivariate environmental space occupied (data available at https://figshare.com/articles/_/8175158). This data set was compared with the available conditions worldwide, in native and non-native ranges, as well as with the species' physiological tolerance limits. In order to evaluate if combining information about these physiological tolerance limits could improve the prediction of *T. scripta*'s potential invasiveness, we simulated again its potential distribution with VIRTUALSPECIES, this time based on the response curves previously estimated and the higher resolution surfaces built for Mexico.

## RESULTS

In accordance with the climatic conditions occupied by *T. scripta* along the two PCA axes (Fig. 2), Schoener's *D* similarity measure suggests that the niche (the native environmental conditions) has been, at least, partially retained ($D = 0.301$) in the invaded range. This observed niche overlap between the native and non-native niches did not deviate from random expectations (*niche similarity test*, Native to Non-native: $P = 0.029$; Non-native to Native: $P = 0.030$).

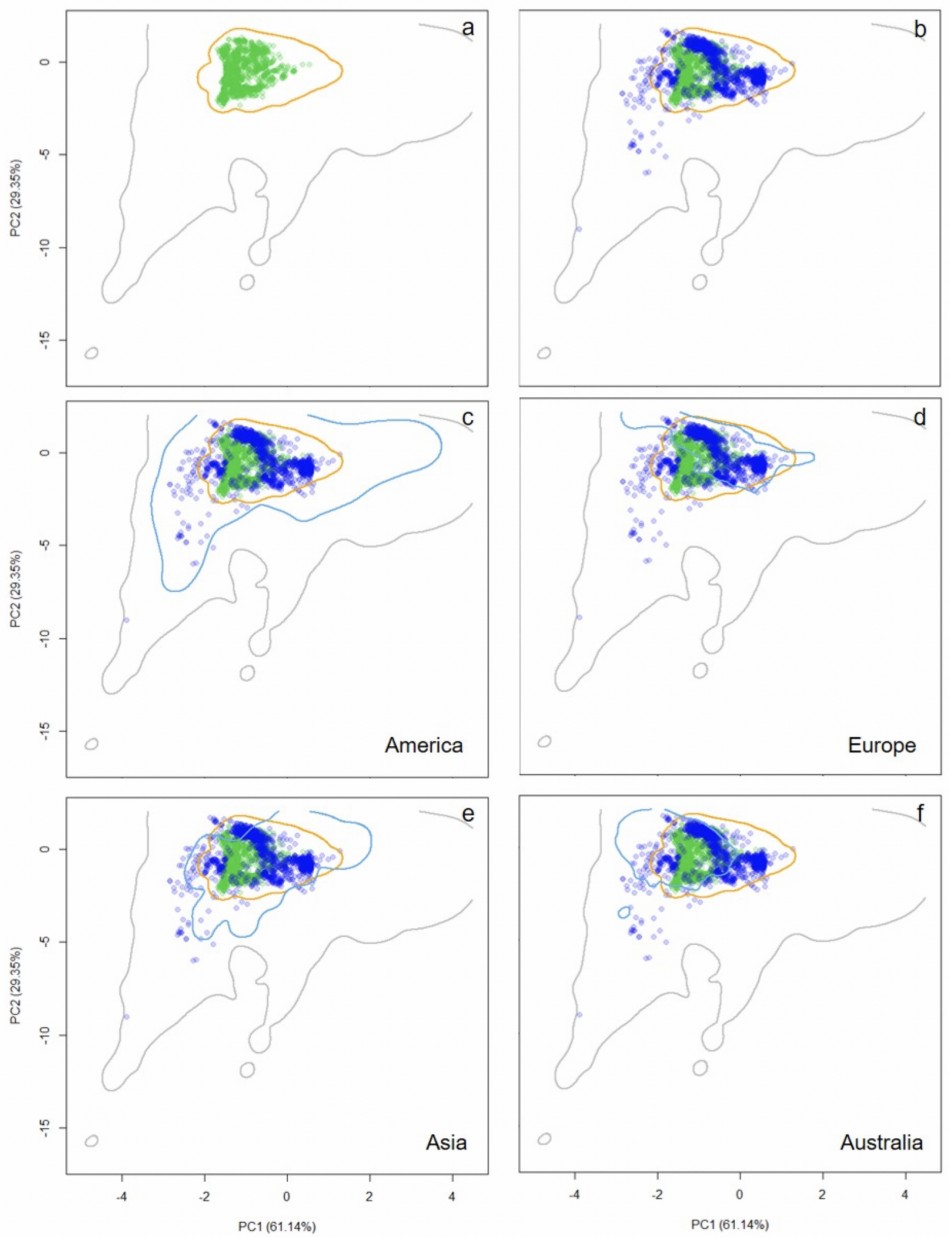

**Figure 2** **Distribution of occurrence records of *Trachemys scripta* along the ecological space.** The distribution is represented by the first two principal components of the Principal Components Analysis. The first two principal components accounted for 90.49% of the variation in the data, including mean annual temperature, mean temperature of warmest quarter, mean temperature of coldest quarter, annual precipitation, precipitation of driest month, and mean annual radiation. Contours delimit the range of climatic conditions available around the world (grey line) and at the native range (orange line). Green dots are *T. scripta* occurrence records at its native range (A) and blue dots outside its native range (B); occurrence densities are depicted by the colour darkness of the dots. Blue contours delimit the range of climatic conditions available in America (C), Europe (D), Asia (E), and Australia (F), where *T. scripta* has successfully invaded (graphics were performed in R).

The CPCA results showed that the best model that explained the differences between the structure of the climatic matrices of the native and non-native realized niches was the one with dissimilar eigenvectors and eigenvalues (AIC = 42.0; Table S2); that is, matrices have different shape, orientation and size, indicating that even when there are similar climatic conditions at both the native and non-native ranges, non-native individuals of *T. scripta* occupy the available environment in a different fashion. In fact, comparing *T. scripta*'s native realized niche with its non-native one revealed incomplete niche stability (70%), while there was evidence of expansion (30%) into climates available worldwide (Fig. 2).

Results of the six-dimensional climatic space analysis showed that the total hypervolume overlaps by 29% and that the non-native realized niche is 28.5% greater than the native one (Fig. S1). The contribution of the climatic variables to the hypervolume differed between range distributions: regarding the native, mean temperature of warmest quarter was the most important variable, whereas for the non-native it was annual precipitation (Table S3).

The characterization of the fundamental niche (temperature tolerance limits for the different behaviours) of *T. scripta* are depicted in Fig. 3A. Notably, our results comparing temperature density curves between native and non-native realized niche ranges (e and f in Fig. 1, respectively) revealed that *T. scripta* does not use its entire range of temperature physiological tolerances (density curves for native: 5.3–23.7 °C and non-native: 1.7–28.4 °C realized niche ranges; Fig. 3B), namely there are unoccupied areas of higher temperatures within its optimal tolerance limits, while occurrence records are biased towards lower temperatures. Moreover, whether considering the mean temperature of the warmest quarter (Fig. 3C) or the coldest quarter (Fig. 3D), it occupies a wider range of temperatures along its non-native distribution. Results of the response curves based on the fundamental niche and distance to freshwater bodies for worldwide comparisons showed that *T. scripta* occupies analog and non-analog conditions between its native and invaded ranges, both within its tolerance limits, although a few occurrence points fall outside of the limits of its fundamental variables (Figs. 4 and 5). Results also showed suitable climatic conditions, but with no exotic occurrence records, in America, Europe, Asia and, to a lesser extent, Australia (Figs. 4 and 5). Finally, the climatic suitability obtained for Mexico exhibits widespread areas with high potential risk of invasion by *T. scripta* (Fig. 6).

## DISCUSSION

Based on our model to evaluate environmental conditions and species features, we show how taking into account both the realized and the fundamental niches (i.e., physiological tolerance limits) for predicting niche changes in novel or non-equilibrium contexts, such as invasions, can improve our understanding about ecological aspects and geographic range boundaries in current and potential invasions (Fig. 1). By considering the temperature physiological tolerances evaluated, we were able to identify that *Trachemys scripta* at its invaded ranges occupies a wider subset of its fundamental niche, which enabled us to ascertain that the species observed niche shift corresponded not to a real change but to a different subset of its fundamental niche (niche unfilling). Our results also show unoccupied areas that have suitable climatic conditions both around the world (Figs. 4 and 5) and regionally within Mexico (Fig. 6) where the slider turtle can potentially invade.

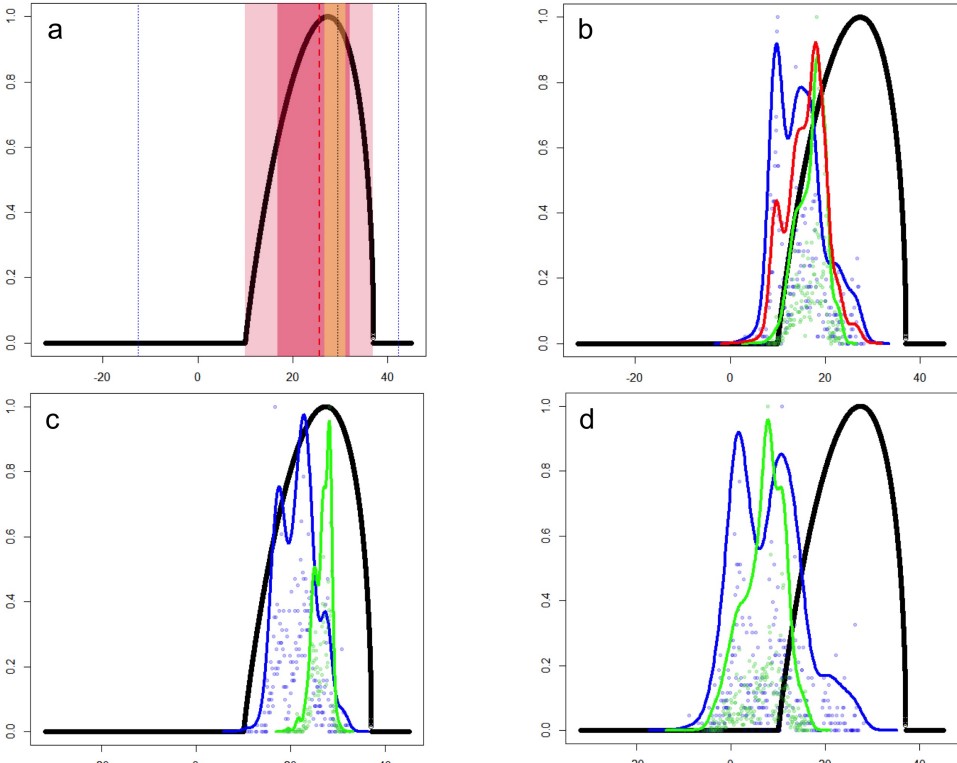

**Figure 3** **Temperature tolerance curves of *Trachemys scripta*'s fundamental niche.** (A) Temperature tolerance limits for different behaviors based on literature reports are depicted: optimum temperature for eggs incubation (29.5 °C) and daily activity (25.6 °C) are represented by black and red dotted lines, respectively; ranges for feeding (16.8–32.1 °C), normal activity (10–37 °C), and basking (26.7–31.2 °C) are shaded in dark-pink, light-pink, and yellow, respectively. Minimum and maximum tolerances before death (−12.6–42.3 °C) are represented with blue dotted lines. The thick black line represents the curve encompassing the temperature tolerances estimated in laboratory trials for all the behaviors considered. (B) Comparison between the optimal temperature tolerances (thick black line) and the mean annual temperature (BIO01) ranges actually occupied by the slider turtles at their native (green line) and non-native (blue line) ranges. The red line represents the mean of the total temperature range occupied by *T. scripta* around the World (i.e., native + non-native ranges). Occurrence probabilities are shown by dots for native (green) and non-native (blue) records. (C) and (D) depict the comparison between the optimal temperature tolerances and the occupied ranges based on the mean temperature of the warmest quarter (BIO10) and the coldest quarter (BIO11), respectively. *x*-axis: temperature (degrees Celsius); *y*-axis: occurrence likelihood (scale 0 to 1) based on the cumulative frequency of records occurring at a particular temperature (graphics were performed in R).

### *Trachemys scripta* unfilled realized and fundamental niches

The approach we followed to define *T. scripta*'s fundamental niche, based on temperature physiological tolerances and availability of freshwater bodies, both conditions essential for the persistence of its populations, revealed it has a wide environmental range with favourable climatic conditions (within its tolerance limits). This result is consistent with earlier works suggesting that *T. scripta* has a broad environmental tolerance, particularly regarding temperature ranges (*Rödder et al., 2009*; *Kikillus, Hare & Hartley, 2010*; *Masin et al., 2014*); remarkably, what had not yet been demonstrated is that such broad tolerance does

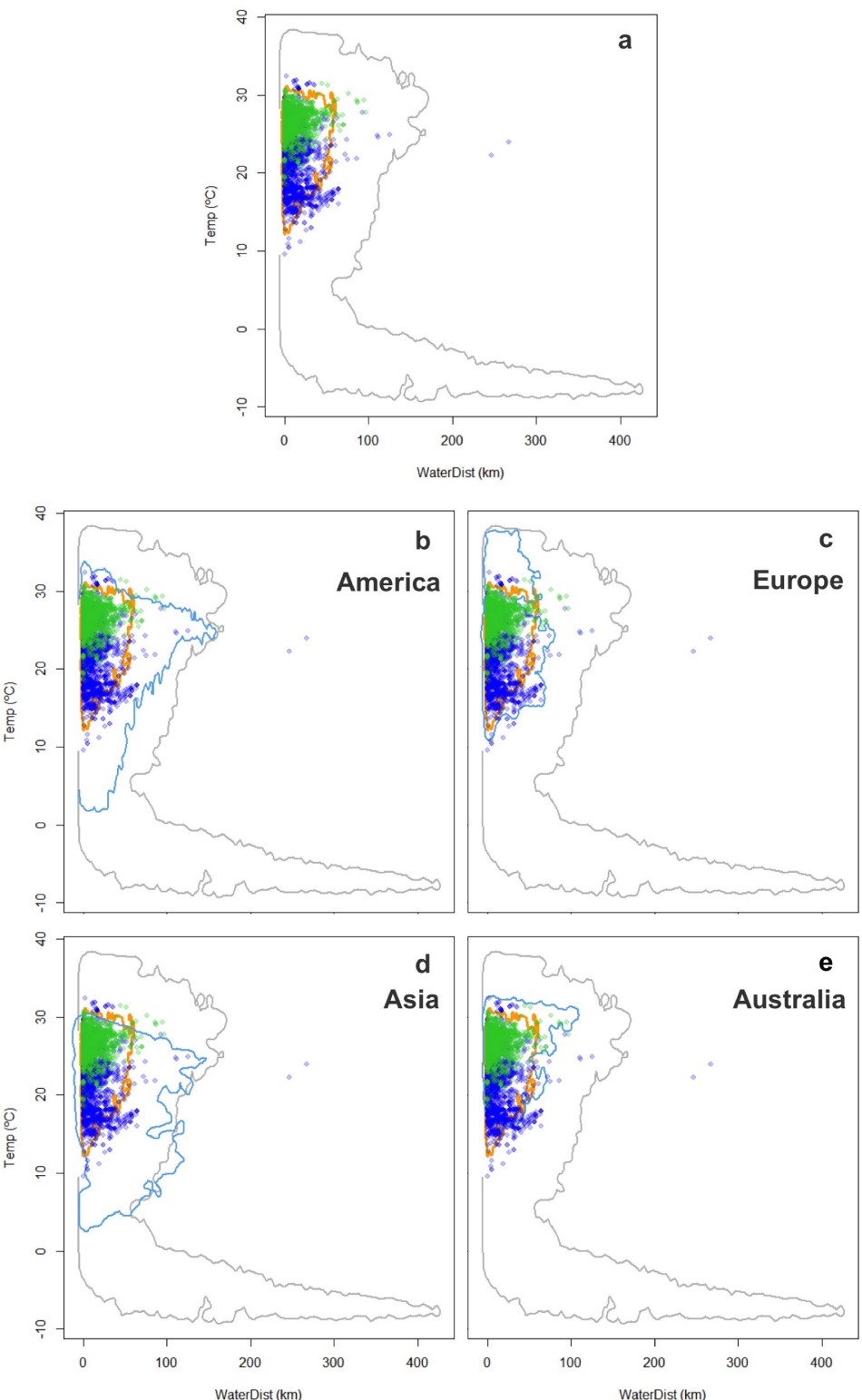

**Figure 4 Worldwide distribution of *Trachemys scripta* along the ecological space I.** The ecological space is represented by the mean temperature of the warmest quarter (BIO10) and the Euclidian distance to the nearest freshwater body. 

**Figure 4 (...continued)**
Contours delimit the range of the environmental conditions available at the native range (orange line) and around the world (grey line). Green dots are *T. scripta* occurrence records at its native range and blue dots outside its native range (A); occurrence densities are depicted by the colour darkness of the dots. Blue contours delimit the range of climatic conditions available in America (B), Europe (C), Asia (D) and Australia (E), where *T. scripta* has successfully invaded (graphics were performed in R).

encompass a fragment of the range of temperature optimums it requires for performing key behaviors (egg incubation, hatching, growth, basking and activity periods). Accordingly, we consider that the climatic data we used to obtain the ranges of the fundamental niche actually occupied by *T. scripta* allowed us to build a robust model encompassing the environmental conditions required for all these behaviors to occur (i.e., the species' fundamental niche).

Indeed, when compared with the range that *T. scripta* actually occupies, we evidence that it is not experiencing the entire range of optimal conditions along its native distribution; specifically when compared based on the warmest conditions, where suitable climate is unoccupied along its higher spectrum of tolerances (Fig. 3). The latter might reflect biotic features (absent from the climatic models), namely the fact that the distribution of its congener species, *Trachemys cataspila*, starts at the southern limit of the slider turtle's, occupying warmer environmental conditions; these two species do not occur sympatrically, likely due to competition (*Seidel, 2002*). Interestingly, we also found that although non-native areas include similar temperature ranges comparatively with the native ones, non-native individuals have invaded environments that are both warmer and colder than its native range, as documented by *Ficetola, Thuiller & Padoa-Schioppa (2009)* and *Rodrigues, Coelho & Ribeiro (2018)*; however, we show that this has happened without deviating from the turtle's tolerance ranges. This could result from elimination of dispersal barriers and of potential competitors, as shown by studies with different taxa where niche shifts (unfilling and expansion) have been associated with biotic interactions and dispersal limitations, which prevent the species from colonizing the full extent of the available (suitable) conditions (*Hargreaves, Samis & Eckert, 2014*; *Tingley et al., 2014*; *Tingley et al., 2016*; *Strubbe, Beauchard & Matthysen, 2015*); or, conversely, as factors that facilitate the establishment of invasive species, exhibiting where they can potentially be found (*Heikkinen et al., 2007*; *Tingley, Phillips & Shine, 2011*; *Giannini et al., 2013*). Our finding that *T. scripta* is also reaching lower temperatures along its invaded distribution could be associated with the trade of the species as a pet (*Burger, 2009*), where repeated introductions might be forcing the species to occupy areas that are not as optimal (*Ficetola, Thuiller & Padoa-Schioppa, 2009*; *Zhu, Li & Zhao, 2017*). For instance, *Ficetola, Thuiller & Padoa-Schioppa (2009)* recorded *T. scripta* individuals in northern Europe, but where conditions are likely too harsh for successful breeding.

The evaluation of native and non-native distributions provides key information in order to decipher if observed shifts might simply indicate different portions of the fundamental niche occupied by the species (*Araújo & Peterson, 2012*), or suggest an adaptation process to new environmental conditions (*Hill, Chown & Hoffmann, 2013*). When we contrasted *T. scripta*'s native and non-native realized niches along the environmental space, we

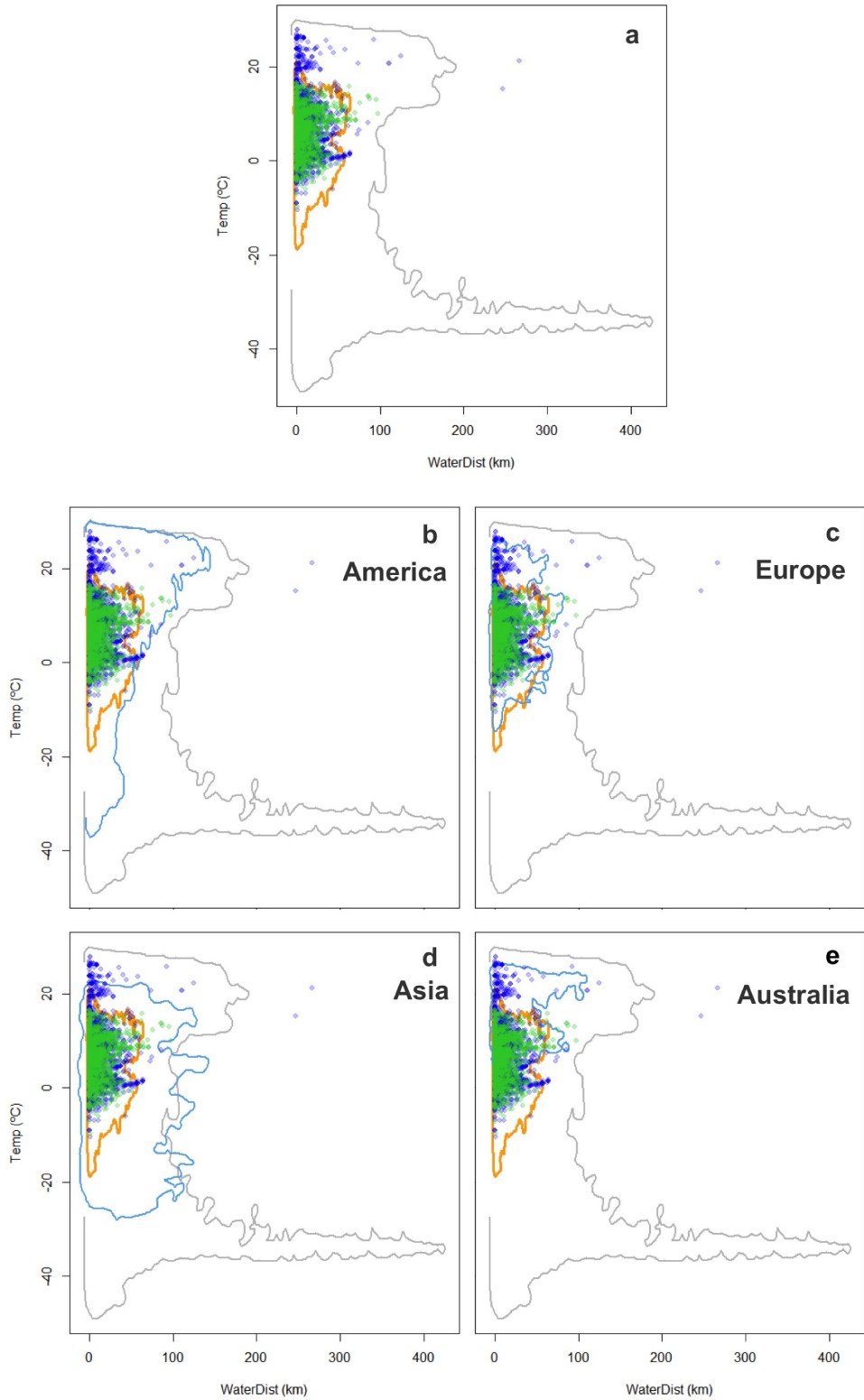

**Figure 5  Worldwide distribution of *Trachemys scripta* along the ecological space II.** The ecological space is represented by the mean temperature of the coldest quarter (BIO11) and the Euclidian distance to the nearest freshwater body. (continued on next page...)

**Figure 5 (…continued)**
Contours delimit the range of the environmental conditions available at the native range (orange line) and around the world (grey line). Green dots are *T. scripta* occurrence records at its native range and blue dots outside its native range (A); occurrence densities are depicted by the colour darkness of the dots. Blue contours delimit the range of climatic conditions available in America (B), Europe (C), Asia (D) and Australia (E), where *T. scripta* has successfully invaded (graphics were performed in R).

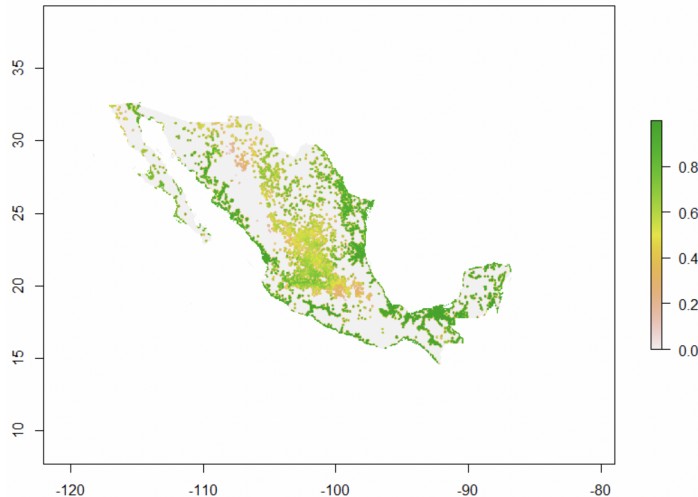

**Figure 6**  *Trachemys scripta*'s **potential environmental suitability across Mexico.** Map for Mexico drawn with Virtualspecies (*Leroy et al., 2015*), showing the areas of climatic suitability for *T. scripta* based on temperature tolerance curves for different behaviors (eggs incubation, feeding, basking, daily activity) and Euclidian distances to the nearest freshwater body (the species' fundamental niche).

found a shift in shape, orientation and size (climatic matrices). In fact, the hypervolume analysis results suggest that the introduction of *T. scripta* into new areas has facilitated the use of available climates in a different way. For instance, while our findings show that temperature is the most significant variable determining the areas that are occupied by the turtle in its native range, it is precipitation at its non-native range. We acknowledge that the hypervolume method's assumptions have been criticized (*Qiao et al., 2017*), nonetheless our results are consistent with the fact that the species always occupies a set of environmental conditions within its physiological tolerance limits and encompassing both analog and non-analog conditions. Furthermore, our results hence support the hypothesis about invasive species not in equilibrium with the environment at their native range (*Elith, Kearney & Phillips, 2010*; *Owens et al., 2013*), and ratify the importance of fully considering the species' capacity to use a wider climate gradient on both space and time (*Broennimann & Guisan, 2008*; *Václavík & Meentemeyer, 2012*; *Tingley et al., 2014*). Accordingly, we show that the slider turtle's documented niche expansion (*Li et al., 2014*; *Rodrigues et al., 2016*) in fact represents a different subset of the species' fundamental niche.

## More complex ways when defining potential invasion risk

Notably, our results show both worldwide suitable conditions (in America, Europe, Asia, Australia) as well as regional areas (Mexico) that have not yet been invaded despite encompassing the range of climatic conditions where *T. scripta* has successfully invaded (Figs. 4–6). In addition to the relationship between the distribution of organisms and climate availability, there are more complex approaches to define potential invasion risks (*Gallien et al., 2010*; *Qiao et al., 2017*), which could also explain why *T. scripta* does not occupy the total area with suitable environment in its non-native (and neither native) distribution. *Václavík & Meentemeyer (2012)* have pointed out the need to consider the stage of the invasion to avoid underestimation of habitats at risk of invasion. The history of invasiveness of *T. scripta* is relatively recent: it has been transported globally since the 1950s as one of the most popular turtle pets (*Burger, 2009*), facilitating its introduction into new habitats. Considering the long generation times of these turtles, it is highly likely that *T. scripta* can continue spreading and colonizing new portions of its fundamental niche, which is currently not being fully occupied, allowing it to respond favourably under a climate change scenario (*Hellmann et al., 2008*; *Cosner, 2014*; *Early & Sax, 2014*).

Ecological niche models need to be taken with caution when used to forecast species invasions and their response to environmental change, as it has frequently been shown they may underestimate the potential spread of invasive species (see *Parravicini et al., 2015*; *Merow et al., 2017*). Occurrence data from within and outside of a species native range have been amply used in studies evaluating invasion risk and niche change in exotic species (*Guisan et al., 2014*; *Li et al., 2014*). Studies have, nonetheless, shown that geographic predictions should be used carefully, especially when species are experiencing niche shifts, because it can lead to underestimation of the potential invasion risk. We believe one crucial condition to evaluate invasion risk is having the most accurate fundamental niche possible of the species in question. Biotic interactions also play an important role during the invasion process and, therefore, are key in defining the range that an invasive species can or cannot occupy. Indeed, although the slider turtle has been able to establish widely, its success has not been ubiquitously identical. One likely reason for the latter is that invasions are species- and case-specific, mainly because it involves complex biotic interactions (i.e., with competitors or predators) not easily detected by environmental modeling approaches. Thus, although still unresolved (see *Wisz et al., 2013*; *Cunningham et al., 2016*), it is essential to find means to include biotic interactions when ENM predictions are used to evaluate invasion risk. Also, for an accurate invasiveness assessment, the distinction between colonizing areas where the species can establish and reproduce from those where it can only survive needs be considered; for instance, by comparing predictions with and without those records. In our case, we consider that *T. scripta* 's observed non-native range is an adequate proxy of its invasive potential, given that it can survive for decades in areas outside its breeding requirements (*Ficetola, Thuiller & Padoa-Schioppa, 2009*) and that the observed non-native ranges do not deviate from its fundamental niche. Finally, spatial heterogeneity in environmental conditions is another challenge to consider when searching for better invasion risk models. Incorporating landscape genetics methods (*Balkenhol et al., 2013*) could aid in detecting barriers or corridors embedded in the geographic space,

ultimately helping improve predictions associated with the expansion range of invasive species (*Tingley et al., 2013*).

## CONCLUSIONS

Our approach (Fig. 1) emphasizes the importance of considering key information regarding the survival of species to detect areas that have suitable conditions, areas that could therefore be potential for invasion. It highlights the need to incorporate links between fundamental niche information, namely functional traits and species constraints that influence the survival and spread of organisms to new habitats, into ENM, to improve our current ability to predict potential invasions. It can be applied with different species for which data pertaining their fundamental niche exists. As we show, even if information about a species' functional traits is not based on biophysical models or experimental trials (sensu *Kearney & Porter, 2009*), combining temperature physiological limits (or other physiological constraint) for different vital activities can provide accurate approximations to derive a mechanistic formulation of its fundamental niche. Furthermore, we show that incorporating response functions analyses and comparing empirical and simulated potential distributions provide a key step to understand the processes limiting species' ranges, as well as to predict and accurately describe niche changes and range shifts. Although our results did not exhibit differences in analog and non-analog conditions between the native and invaded ranges, our approach permits to detect such differences where they exist. Undoubtedly, this allows for more robust predictions for invasive species that, as has been evidenced more and more often, are not in environmental equilibrium (*Kearney & Porter, 2009*; *Elith, Kearney & Phillips, 2010*). Indeed, our findings about climatic suitability regions for *T. scripta*, as in our example for Mexico, exhibit a more precise identification of those areas that have the highest potential for a successful introduction of this turtle. Moreover, considering the continuous movement of the slider turtle into new environments via human introductions, and the wide range of environmental conditions suitable and not yet occupied by this turtle, the risk of invasion is rather significant. Integrating other approaches, such as stages of invasion, future climate change, environmental heterogeneity, and biotic interactions to evaluate this and other species invasion processes is a major challenge, yet necessary in order to achieve more reliable models for the management and control of invasive species.

## ACKNOWLEDGEMENTS

Special thanks to Juan Fornoni and Enrique Martínez Meyer for helpful discussions throughout the project. We thank the people from Juan Parra's lab at the Universidad de Antioquia for their assistance to this work during Sayra Espindola scientific visit. This paper constitutes a partial fulfilment of the Graduate Program in Biological Sciences of the National Autonomous University of Mexico (UNAM).

### Funding

This work was supported by Consejo Nacional de Ciencia y Tecnología (Conacyt 237228), and with a Sayra Espindola's scholarship and financial support (CONACyT CVU 364630/Reg. becario 245447) and Scholarship Program for Postgraduate Studies (PAEP). This article was finished while Ella Vázquez-Domínguez was on sabbatical at the American Museum of Natural History (DGAPA/PASPA 20160609). The funders had no role in study design, data collection and analysis, decision to publish, or preparation of the manuscript.

### Grant Disclosures

The following grant information was disclosed by the authors:
Consejo Nacional de Ciencia y Tecnología: 237228.
Sayra Espindola's scholarship and financial support: 364630.
Scholarship Program for Postgraduate Studies.
American Museum of Natural History: 20160609.

### Competing Interests

The authors declare there are no competing interests.

### Author Contributions

- Sayra Espindola conceived and designed the experiments, performed the experiments, analyzed the data, prepared figures and/or tables, authored or reviewed drafts of the paper, approved the final draft.
- Juan L. Parra conceived and designed the experiments, performed the experiments, analyzed the data, contributed reagents/materials/analysis tools, prepared figures and/or tables, authored or reviewed drafts of the paper, approved the final draft.
- Ella Vázquez-Domínguez conceived and designed the experiments, contributed reagents/materials/analysis tools, prepared figures and/or tables, authored or reviewed drafts of the paper, approved the final draft.

### Data Availability

Data is available at Figshare: Vázquez-Domínguez, Ella; Espindola, Sayra; Parra, Juan (2019): DataTrachemysScripta.csv. figshare. Dataset. https://doi.org/10.6084/m9.figshare.8175158.v1.

### Supplemental Information

Supplemental information for this article can be found online at http://dx.doi.org/10.7717/peerj.7923#supplemental-information.

## PeerJ

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
