# Peer review of "Fundamental niche unfilling and potential invasion risk of the slider turtle Trachemys scripta"

_PeerJ, doi:10.7717/peerj.7923_

## Round 0.1 · original submission · Major Revisions

Both the reviewers and myself believe this paper is interesting and provide important information about an invasive species. I agree with all of the comments made by both reviewers so I ask the authors to pay close attention and resolve all of the queries, especially the use of the terminology which both reviewers find not completely appropriate.

Reviewer 1 ·

Basic reporting

The manuscript from the authors presents a study about climatic niche differences between native and non-native populations of the slider turtle Trachemys scripta.

I found the manuscript very well written and clear and the methods used. The references were sufficient and well cited. The figures need to be substantially improved due to their poor quality.

The authors consider that the fundamental niche can be defined from physiological limits (e.g., maximum and minimum critical temperatures) However, this is not an accurate definition of the fundamental niche (sensu Hutchinson 1957). Although this has been a pervasive confusion in the literature for years it should be clear that the fundamental niche definition was erected from a demographic basis (see Hutchinson 1957 and Holt 1997). In other words, the fundamental niche is the environmental conditions that influence birth and death rates on a population allowing its survival without immigration. I would like to see a more deep explanation from the authors about this issue and why they decide to use physiological proxies operating at the individual level to operationalize the concept of fundamental niches.

References

Hutchinson, G.E. (1957) Concluding Remarks. Cold Spring Harbor Symposia on Quantitative Biology, 22, 415-427

Holt, R.D (2009). Bringing the Hutchinsonian niche into the 21st century: Ecological and evolutionary perspectives. PNAS 106: 19659–19665.

Experimental design

The authors compiled a lot of physiological information from this species and they use this data to generate thermal tolerance curves and predict potential distribution across the world.

The authors did not include a proper comparison of analog or non-analog climate conditions between native and non-native areas. This comparison likely is crucial to establish whether native and non-native areas are climatically different and can be used to establish whether invasive T. scripta populations are occupying non-analog climates in other areas. Although there is a full set of methods to test this (see for example Elith et al. 2010; Mesgaran et al. 2014, etc.), it is strange the lack of this in this work.

References

Elith et al. (2010). The art of modelling range-shifting species. Methods in Ecology and Evolution 1: 330-342.

Mesgaran et al. (2014). Here be dragons: a tool for quantifying novelty due to covariate range and correlation change when projecting species distribution models. Diversity and Distributions 20: 1147-1159

Validity of the findings

The authors presented two kinds of results about niche comparisons from native and non-native ranges. By one hand, null model tests revealed that there are no differences between native and non-native realized niches (lines 289-293). Randomization tests conducted by the authors suggest that even minor niche differences between native and non-native areas cannot be different from a random point selection through the entire background area. By another hand, their PCA matrix comparisons revealed a difference in climate structure from native and non-native areas (line 294-302). However, these PCA comparisons did not included a randomization test, which is crucial in this kind of exercises. Furthermore, thermal tolerance curves from physiological data suggest that native and non-native areas (see fig. 3) are included within the full curve from temperature (i.e., the "fundamental niche" as authors defined previously).

It is hard to see that native and non-native niches are different as authors concluded based on the data at hand and the statistical analyses. The data and tests used here suggest to me that T. scripta is occupying a large portion of its "fundamental niche" in non-native areas likely due to the elimination of dispersal barriers and/or elimination of potential competitors/predators.

Additional comments

The manuscript from the authors presents a study about climatic niche differences between native and non-native populations of the slider turtle Trachemys scripta. I found this study very interesting because compile a lot of physiological information which is used to generate thermal tolerance curves and predict potential distributions in geographical space. However, the results suggest that there are no differences in climatic niches in native and non-native areas and a proper comparison of analog and non-analog climates across areas is recommended.

Reviewer 2 ·

Basic reporting

I have two comments in this section. One is related to figure 3. The part (a) of the figure is smaller than the other ones, please correct it.

My second comment is related to English. I would suggest the authors ask a Native speaker for proof-reading of the entire manuscript, especially after the changes.

Experimental design

I found the methods employed appropriate, in general, but I have the following comments.

Lines 138-159. All these sentences are a good summary of the entire approach used here and it is not bad. However, I think the authors should consider moving this part to the first part of the methods. The introduction is already big and not easy to read because of the diversity of information and the structure in which it was written.

Line 173. 5 km is around 2.5 min. Given the resolution of the raster layers for global analyses, this means that not even duplicates where removed. A larger distance must be used for global analyses to avoid duplicate records. I consider that increasing the distance may not change the findings and conclusion of this work. However, if the authors decide not to improve this part, an explanation of what potential effects can be expected from this caveat should be mentioned in the discussion.

Line 180. In line 275 the authors also talk about BIO10 and BIO11, I consider that those variables should be mentioned here as well.

Lines 192, 199. I disagree in that a distance layer defines a characteristic of the fundamental niche. A population of this species is going to be OK as long as the water body in which it exists has appropriate conditions and individuals do not have to move. Population size can be relatively stable once it reaches its carrying capacity. Whether or not there is another water body close or far won't affect such population. Please argument better why the authors included this variable.

Line 206. How the probability value was calculated? Why weighting is necessary? Conditions are available whether or not the species is using them and depends on other factors especially related to history and mobility? Why not defining accessible areas instead of doing these weighting? Why all turtles and not only the ones that are closely related to the one of interest? Please clarify in the text, the readers will find these explanations very useful.

Line 250. If the distance-derived variable is included, the assumption that the response of the species to this variable is similar to responses to temperature and precipitation is big. Describe why this assumption is not problematic.

Validity of the findings

The findings are interesting and from my point of view, valid. Please see some comments for the results and discussion sections below.

Line 290. I assume this result (the niche has been, at least, partially retained (D=0.287) in the invaded range) is concerning the native environmental conditions. Please be specific.

Line 294. This sentence is part of methods, please move it to that section.

Line 325. Please explain better this part or reword the sentence. How is it possible to invade an area and do not occupy it?

Line 331. How are the authors evaluating niche dynamics? What do the authors mean by that? Dynamics of what niche (fundamental, potential, realized)?

Line 352. Based on figure 3, it looks like BIO01 and BIO11 are not as good variables as the authors suggest. Occupied environmental conditions characterized by BIO1 and BIO11 do not match with the fundamental niche as good as BIO10 does. Please discuss these ideas in the text. What the authors can suggest about variable selection processes in ENM, for instance?

Line 387. I am not sure what the authors are trying to say here. Species fundamental ecological niches are defined by the limits of their tolerance to variables. How come a species can occupy distinct niches within its tolerance ranges? What niches are the authors referring to? Please clarify

Line 399. Do the authors have any evidence of the species going to invade these areas?

Line 412. As from I understood, the authors did not consider what invasive records correspond to established populations or individuals that are only surviving. Please discuss what implications this fact could have in the results found in this work.

Line 432. This is correct but it would be nice to know if the authors have ideas of other potential solutions. What the authors can say about combining records from native and invaded areas (established populations) to create better ENMs for invasive species? Do the authors think it can be an option for species where fundamental niches are unknown?

Additional comments

The main idea behind the manuscript entitled “Fundamental niche unfilling and potential invasion risk of the slider turtle Trachemys scripta” is interesting and worth to explore. I consider this is an acceptable contribution; however changes are necessary to improve it. My general comments are listed below.

- The manuscript is long and considering the analyses performed and the information presented it can be improved by avoiding some information that may not be that relevant. For instance, introduction and discussion are difficult to follow because of its length.

- The authors say they developed a conceptual model and I disagree with that. What the authors did was to use distinct tools and analyses to investigate a specific question. Therefore, they only used an approach that can be modified in many different ways in other studies. What the authors did in this study do not allow to compare if this approach is better than others than can be followed to perform comparable analyses; therefore I consider the authors should refer to their set of analyses as an approach and not a conceptual model.

- The use of terminology in the manuscript is not completely appropriate. Although the authors made an effort in including a Glossary, they are not consistent with terms like niche and niche shift. The distinct types of niches are not used properly in all their mentions. I would suggest the use of three types of niches: fundamental, existent, and realized (see the Book of Peterson et al. 2011).

Especial comments:
- Line 72. "... however, an important limitation when they are used for range predictions is the assumption made that the climatic niche between the native and the non-native (invaded or exotic) range is conserved across space and time". This is not at all a limitation, on the contrary, this is the base to make these methods applicable to such questions. This paper is another proof of the usefulness of this assumption and the fact that the limitation is not the assumption but the way in which ecological niches are catheterized.

- Line 121. Determine is not an appropriate verb here, perhaps identify works better for what you are doing.

- The paragraph in lines 107-123 may be better located after the paragraph that is following it in the current version of the manuscript.

---

## Round 0.2 · accepted · Accept

The authors modified the manuscript as all reviewers had suggested previously and I agree with the second round of reviews in that the manuscript has improved in the use of concepts and also with new analysis.

Reviewer 1 ·

Basic reporting

The authors present a revised version of this manuscript with clarification about definitions and proper comparisons of analog and non-analog conditions. The issue of the fundamental niche definition was revised based on a review of classical papers from Hutchinson. Although I disagree with the Soberón & Arroyo-Peña (2017) about the fundamental niche definition, which it should be not related to physiological thermal limits, I think the authors provided a relatively decent explanation about how this data can be used as a proxy. Anyway, they discussed and explained this in the main text.

The authors also corrected the confusions about niche differences in native and invaded ranges and conducted a proper comparison in the environmental space. The comparison of the structure of the covariance matrix in ecological space is a novel approach here to evaluate whether analog and non-analog conditions exist across different geographical settings. I think this part resolves the problem identified in the first version.

Experimental design

The experimental design based on the new analyses included is ok.

Validity of the findings

I think the authors rewording the text to avoid confusions between geography (e.g., native and invaded range) and ecological space (e.g., analog vs. non-analog climates). I'm ok with these changes.

Additional comments

I think the authors present a very interesting study with new statistical approaches to compare realized niches across geography in invasion ecology.